# Unifying Linear-Time Attention via Latent Probabilistic Modelling

**Rares Dolga**                                                   *rares.dolga.16@ucl.ac.uk*
*University College London, AI Centre*
*UiPath*

**Lucas Maystre**                                                 *lucas.maystre@uipath.com*
*UiPath*

**Marius Cobzarenco**                                            *marius.cobzarenco@uipath.com*
*UiPath*

**David Barber**                                                  *david.barber@uipath.com*
*University College London, AI Centre*
*UiPath*

**Reviewed on OpenReview:** *https://openreview.net/forum?id=TDFIjR7ynG*

## Abstract

Transformers have achieved state-of-the-art results across a range of domains, but their quadratic attention mechanism poses significant challenges for long-sequence modelling. Recent efforts to design linear-time attention mechanisms have yielded more scalable alternatives, yet often at the cost of performance, particularly on discrete data such as language. In this work, we revisit linear attention through the lens of probabilistic graphical models. We first show that standard linear attention can be interpreted as an undirected latent variable model, revealing a key limitation: the absence of directionality. To address this, we propose a novel *directed* parameterisation of linear attention that introduces an asymmetric structure, enabling an interpretation aligned with the causal and sequential nature of language. Our formulation integrates global latent-variable attention with local standard attention in a fully probabilistic framework. Additionally, we introduce a recurrent parameterisation of queries and keys that avoids reliance on relative positional encodings, often incompatible with linear attention. Experiments on language modelling benchmarks demonstrate that our model achieves competitive performance with standard attention and outperforms existing linear attention variants.

## 1 Introduction

Transformers have become a cornerstone of modern deep learning, with attention mechanisms at the heart of their success. Despite their strong performance across a wide range of tasks, the *quadratic* time complexity of standard attention imposes efficiency challenges when scaled to extremely long sequences. This limitation has motivated extensive research into *linear-time* attention mechanisms. While several linear variants have been proposed (Katharopoulos et al., 2020; Lin et al., 2021), most fall short of the performance achieved by standard Transformers, particularly in language modelling.

Recent work highlights the importance of gating mechanisms for improving linear attention performance on discrete data such as text (Yang et al., 2024). However, these approaches still build on the classical formulation of linear attention, which approximates softmax attention by expressing it as an inner product of kernel feature maps and leveraging associativity of matrix multiplication. Although the standard attention mechanism can be viewed as a probability distribution, the linear formulation lacks a probabilistic foundation,

and it remains an open question whether stronger linear mechanisms can be developed by drawing on probabilistic principles.

In parallel, many hybrid models combine linear global attention with local standard attention (Hua et al., 2022). However, these methods often fail to properly normalize attention weights across the sequence, which can introduce biases, such as putting more weight on the local attention.

In this work, we revisit the foundations of linear attention through the lens of probabilistic graphical models. We first show that standard linear attention can be interpreted as an *undirected latent variable model*. Building on this insight, we propose a novel *directed* parameterisation of linear attention. We will show that although both the undirected and directed models encode the same conditional independence structure, the directed variant enables a generative interpretation, where a token $t$ influences another token $s$ via a latent variable $\ell$. This *asymmetry* aligns with the inherently directional nature of attention and is particularly well-suited for modelling sequential or causal relationships.

Conceptually, our approach can be viewed as clustering tokens around high-level latent concepts, with attention computed via interactions between tokens and these latent variable states. Our framework naturally supports the integration of global and local attention while maintaining a properly normalized probabilistic structure.

**Our main contributions are:**

- We provide a probabilistic interpretation of standard linear attention as an undirected latent variable model.

- Leveraging this insight, we introduce a novel directed parameterisation of linear attention that enables asymmetric modelling.

- We show how to integrate standard local attention with our approach while preserving full normalization of attention weights.

- Finally, we propose a recurrent parameterisation of queries and keys, replacing relative positional encodings (which are incompatible with linear attention), and demonstrate strong performance on language modelling tasks-comparable to standard attention and recent state-of-the-art linear models.

## 2 Background

Viewing attention through a probabilistic lens provides a unified account of both standard and linear attention and clarifies their relationships. Since attention is a probability distribution, standard latent-variable parameterisations are natural. This framing places common variants within approximate probabilistic inference, makes their implicit approximations explicit, and enables the use of classical methods grounded in latent variables.

From this perspective, we propose a directed (unsymmetric) parameterisation of linear attention that is well suited to language. We further develop a fully normalised hybrid that combines this directed variant with sliding-window standard attention and achieves strong empirical performance. These results demonstrate how the probabilistic view yields concrete methodological improvements.

### 2.1 Probabilistic Model for Attention

The attention mechanism takes an input sequence of token vectors $\boldsymbol{X} = \mathrm{vcat}\left(\boldsymbol{x}_1^{\mathsf{T}}, \ldots, \boldsymbol{x}_T^{\mathsf{T}}\right)$; $\boldsymbol{x}_s \in \mathbb{R}^D$ and transforms this to a new sequence according to an input-dependent linear function:

$$\tilde{\boldsymbol{X}} = \mathrm{Attention}(\boldsymbol{X}) \equiv \mathrm{softmax}\left(\boldsymbol{Q}\boldsymbol{K}^{\mathsf{T}} \otimes \boldsymbol{M}\right)\boldsymbol{V} \tag{1}$$

where $\boldsymbol{Q} = \boldsymbol{X}\boldsymbol{W}_q$, $\boldsymbol{K} = \boldsymbol{X}\boldsymbol{W}_k$, $\boldsymbol{V} = \boldsymbol{X}\boldsymbol{W}_v$, $\boldsymbol{M}$ is the causal attention mask or a matrix of ones for bidirectional attention. $\boldsymbol{W}_q, \boldsymbol{W}_k, \boldsymbol{W}_v \in \mathbb{R}^{D \times D'}$.

We can now rewrite the vectorised Equation 1 for each token in the sequence. Although we focus on the causal case, the bidirectional case is similar.

$$\tilde{\boldsymbol{x}}_t = \sum_{s=1}^{t} a_{ts} \boldsymbol{v}_s \tag{2}$$

where $a_{ts}$ are the attention weights defined for the causal case as:

$$a_{ts} = \begin{cases} \dfrac{\exp\left(\boldsymbol{q}_t^\top \boldsymbol{k}_s\right)}{\sum_{s'=1}^{t} \exp\left(\boldsymbol{q}_t^\top \boldsymbol{k}_{s'}\right)} & \text{if } s \le t \\ 0 & \text{otherwise} \end{cases} \tag{3}$$

Since $a_{ts}$ is a positive normalised quantity, we can interpret $a_{ts}$ as the probability $p(s|t)$ of the token (i.e the sequence element) occurring at position $s$ given the token occurring at position $t$. While attention represents learning $p(s|t)$ directly and using this quantity, the joint distribution $p(s, t) = p(s|t)p(t)$ can be represented by the Markov Network in Figure 1.

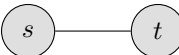

Figure 1: Graphical model for bidirectional attention, $t$ and $s$ are discrete random variables in $\{0, \cdots, T\}$

## 3 Linear Attention as a Graphical Model

In this section, we first show that, under certain assumptions, the vanilla linear attention mechanism (Katharopoulos et al., 2020) can be expressed as an undirected graphical model. Then we introduce Latte, a novel direct parameterisation, which we further extend by combining it with local standard attention. Our work is among the first which combines linear attention and standard attention, such that the weights are fully normalised across the sequence.

### 3.1 Undirected parametrisation

The main disadvantage of attention is the quadratic time complexity which comes from the $\boldsymbol{Q}$, $\boldsymbol{K}$ matrix multiplications in Equation 1. Linear attention avoids the quadratic cost by approximating $\exp\left(\boldsymbol{x}^\top \boldsymbol{y}\right) \approx \phi(\boldsymbol{x})^\top \phi(\boldsymbol{y})$, where $\phi : \mathbb{R}^{D'} \to \mathbb{R}^L$ and making use of the associativity property of matrices.

We look at the expression only for the causal case, the bidirectional being very similar:

$$\tilde{\boldsymbol{x}}_t = \frac{\sum_{s=1}^{t} \phi(\boldsymbol{q}_t)^\top \phi(\boldsymbol{k}_s) \boldsymbol{v}_s}{\sum_{s=1}^{t} \phi(\boldsymbol{q}_t)^\top \phi(\boldsymbol{k}_s)} = \frac{\left[\sum_{s=1}^{t} \boldsymbol{v}_s \phi(\boldsymbol{k}_s)^\top\right] \phi(\boldsymbol{q}_t)}{\phi(\boldsymbol{q}_t)^\top \sum_{s=1}^{t} \phi(\boldsymbol{k}_s)} = \frac{\boldsymbol{S}_t \phi(\boldsymbol{q}_t)}{\phi(\boldsymbol{q}_t)^\top \boldsymbol{z}_t}, \tag{4}$$

where $\boldsymbol{S}_1 = \boldsymbol{v}_1 \phi(\boldsymbol{k}_1)^\top \in \mathbb{R}^{D' \times L}$ and we can then recursively write $\boldsymbol{S}_t = \boldsymbol{S}_{t-1} + \boldsymbol{v}_t \phi(\boldsymbol{k}_t)^\top$ with the normalisation $\boldsymbol{z}_1 = \phi(\boldsymbol{k}_1) \in \mathbb{R}^{D'}$ and $\boldsymbol{z}_t = \boldsymbol{z}_{t-1} + \phi(\boldsymbol{k}_t)$.

The basis function $\phi$ can be any function, but when it is positive, we can assign a probabilistic interpretation for linear attention using the Markov model with a discrete latent variable $l$ with $L$ states as in Figure 2.

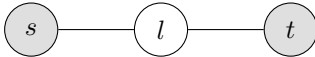

Figure 2: Graphical model for linear attention, $t$, $s$ are discrete random variables and $l$ is a discrete latent variable.

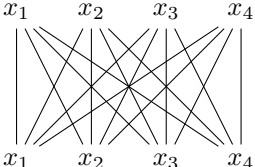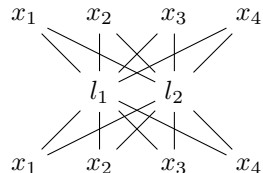

Figure 3: Token-token interaction diagram for non-causal attention of the $p(s|t)$ matrix, see also Lin et al. (2021). (Left) Standard Attention computes all pairwise similarities between the elements of the sequence. (Right) Latte computes only the pairwise similarities between each element of the sequence and each latent state.

In this model we have:

$$a_{ts} = p(s|t) = \frac{p(s,t)}{\sum_{s'=1}^{t} p(s',t)} = \sum_{l=1}^{L} \frac{p(s,l,t)}{\sum_{l'=1}^{L}\sum_{s'=1}^{t} p(s',l',t)} = \sum_{l=1}^{L} \frac{\psi(s,l)\psi(l,t)}{\sum_{l'=1}^{L}\psi(l',t)\sum_{s'=1}^{t}\psi(s',l')} \qquad (5)$$

where we parametrised the joint $p(s,l,t) = \frac{\psi(s,l)\psi(l,t)}{Z}$.

We observe that for $\psi(t,l) = \phi(\boldsymbol{q}_t)_l$, $\psi(s,l) = \phi(\boldsymbol{k}_s)_l$, Equation 5 is equivalent to Equation 4. This shows that for the special case when the basis function $\phi$ is positive, linear attention has a probabilistic interpretation. Additionally, for $L \ll T$, linear attention gives a low rank parametrisation of attention. Nonetheless, this parametrisation is symmetric, which does not reflect the directed nature of language.

## 4 Latte: Directed Parametrisation

Using the connection we have made between linear attention and the undirected latent variable model, we propose our model, Latte. Latte is a novel directed parameterisation for low-rank linear attention as shown in Figure 4. The directed formulation reflects asymmetric dependencies, a property essential for capturing causal or temporal structure and naturally compatible with the flow of information in attention mechanisms. Our model can be seen as performing clustering around the latent states, which can be thought as general learnable concepts such as shapes or colours. While attention performs the normalised similarity between each pair of tokens $\boldsymbol{x}_s$ and $\boldsymbol{x}_t$, our directed latent parametrisation, performs similarity between a token $\boldsymbol{x}_t$ and the latent state $l$.

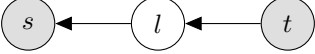

Figure 4: Graphical model for Latte, $l$, $t$ and $s$ are discrete random variables.

Let $l$ be a latent variable with $L$ possible states and $\boldsymbol{x}_1, \ldots, \boldsymbol{x}_T$ a sequence of input tokens. Our model takes the assumption of independence between tokens at time $s$ and $t$: $s \perp\!\!\!\perp t|l$. We define the bidirectional Latte case as:

$$\tilde{\boldsymbol{x}}_t = \sum_{s=1}^{T}\sum_{l=1}^{L} p(s,l|t)\boldsymbol{v}_s = \sum_{l=1}^{L} p(l|t)\sum_{s=1}^{T} p(s|l)\boldsymbol{v}_s \qquad (6)$$

A more intuitive explanation of our model can be found in Figure 3. Similarly, we define the causal case as:

$$\tilde{\boldsymbol{x}}_t = \sum_{s=1}^{t}\sum_{l=1}^{L} p(s,l|t)\boldsymbol{v}_s = \sum_{l=1}^{L} p(l|t)\sum_{s=1}^{t} p(s|l,t)\boldsymbol{v}_s \qquad (7)$$

We note that $p(s|l,t)$ does not imply a quadratic dependency like the standard attention, since we only use $t$ to define normalisation. This becomes clear in Equation 9 which also highlights a key difference between

our model and the linear undirected model in Equation 5.

$$a_{ts} = p(s|t) = \sum_{l=1}^{L} p(s|l,t)p(l|t) \tag{8}$$

$$= \sum_{l=1}^{L} \frac{\psi(s,l)}{\sum_{s=1}^{t} \psi(s,l)} \frac{\psi(l,t)}{\sum_{l'=1}^{L} \psi(l',t)} \tag{9}$$

Similarly to the undirected parametrisation, we define $\psi(l,t) = \phi(\boldsymbol{q}_t)_l = e^{\boldsymbol{x}_t^\top \boldsymbol{w}_l^q}$ and $\psi(s,l) = \phi(\boldsymbol{k}_s)_l = e^{\boldsymbol{x}_s^\top \boldsymbol{w}_l^k}$, which ensures that the potentials are positive. For the bidirectional case, we simply have to replace $t$ in Equation 9 to $T$.

We highlight that the causal case can be written recursively, which enables fast generation at test time. To see this we define the normalisation terms

$$\beta_t \equiv \sum_{j=1}^{L} e^{\boldsymbol{x}_t^\top \boldsymbol{w}_j^q}, \qquad \alpha_{t,l} \equiv \sum_{s=1}^{t} e^{\boldsymbol{x}_s^\top \boldsymbol{w}_l^k} \tag{10}$$

and write the new representation as

$$\tilde{\boldsymbol{x}}_t = \sum_{l=1}^{L} p(l|t) \sum_{s=1}^{t} p(s|l,t) \boldsymbol{v}_s \tag{11}$$

$$= \sum_{l=1}^{L} \frac{e^{\boldsymbol{x}_t^\top \boldsymbol{w}_l^q}}{\beta_t \alpha_{t,l}} \sum_{s=1}^{t} e^{\boldsymbol{x}_s^\top \boldsymbol{w}_l^k} \boldsymbol{v}_s = \sum_{l=1}^{L} \gamma_{t,l} \tilde{v}_{t,l} \tag{12}$$

where

$$\gamma_{t,l} \equiv \frac{e^{\boldsymbol{x}_t^\top \boldsymbol{w}_l^q}}{\beta_t \alpha_{t,l}}, \qquad \tilde{v}_{t,l} \equiv \sum_{s=1}^{t} e^{\boldsymbol{x}_s^\top \boldsymbol{w}_l^k} \boldsymbol{v}_s \tag{13}$$

Since $\tilde{v}_{t,l}$ and $\alpha_{t,l}$ are cumulative sums we can use the recursions

$$\alpha_{t,l} = \alpha_{t-1,l} + e^{\boldsymbol{x}_t^\top \boldsymbol{w}_l^k}, \qquad \tilde{v}_{t,l} = \tilde{v}_{t-1,l} + e^{\boldsymbol{x}_t^\top \boldsymbol{w}_l^k} \boldsymbol{v}_t \tag{14}$$

From Equation 14 it immediately follows that we can calculate $\tilde{\boldsymbol{x}}_{t+1}$ (i.e infer the future token) directly from $\alpha_{t,l}, \beta_t, \tilde{v}_{t,l}$, whereas standard attention requires the full sequence $x_1, \ldots, x_t$. In this sense, Causal Latte is a recurrent model, similar in essence to Recurrent Neural Networks (RNNs) and state space models (SSMs) (Gu et al., 2021; Smith et al., 2022; Fu et al., 2023), see Figure 5.

Our recursive form admits a natural "memory" interpretation: $\tilde{v}_{t,l}$ summarizes past information and is updated once per time step. However, not all linear recurrent architectures fit our probabilistic framework. For example, models such as Mamba(Gu & Dao, 2023) lack the normalization needed to interpret the sequence-mixing operation as a probability distribution, preventing a probabilistic reading of their updates. In contrast, our construction supports low-rank approximations within a coherent probabilistic model (while orthogonal techniques like sparse-attention implementations are outside our present scope).

### 4.1 Hybrid Model

Linear models reduce the time complexity from quadratic to linear, however, their performance lacks behind standard attention (Yang et al., 2024). Our formulation suffers from the same problem. Although it uses latent states to represent global concepts and share long-range information across a sequence, it may not account for local information as effectively as standard attention since it lacks non-linear element-wise comparisons. Therefore combining linear attention with standard attention is natural and has been explored in works such as Hua et al. (2022); De et al. (2024). Different to prior models in our work we maintain a

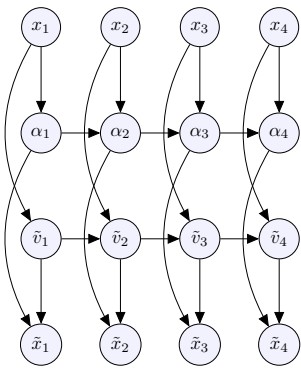

Figure 5: Causal Latte can be written as a recursion in which the variables $\alpha_t = [\alpha_{t,1}, \ldots, \alpha_{t,L}]$ and $\tilde{v}_t = [\tilde{v}_{t,1}, \ldots, \tilde{v}_{t,L}]$ contain all the information required to form the transformed output $\tilde{x}_t$.

properly normalised attention when we combine local and global context by a simple extension of our latent variable model.

We achieve a correctly normalised attention by defining a special latent state, $l = 0$, allocated to standard attention. Then the new model, called Latte Macchiato, is a weighted mixture of standard attention and Causal Latte:

$$\tilde{\boldsymbol{x}}_t = p(l=0|t) \sum_{s=1}^{t} p_0(s|t)\boldsymbol{v}_s \tag{15}$$

$$+ \sum_{l=1}^{L} p(l|t) \sum_{s=1}^{t} p(s|l,t)\boldsymbol{v}_s \tag{16}$$

Here $p_0(s|t) \equiv p(s|l=0,t)$ represents standard attention; in practice to retain computational tractability we use sliding window attention with a window size $w$:

$$p_0(s|t) = \begin{cases} \dfrac{e^{\boldsymbol{q}_t^\mathsf{T}\boldsymbol{k}_s}}{\sum_{s=t-w}^{t} e^{\boldsymbol{q}_t^\mathsf{T}\boldsymbol{k}_s}} & t-w \leq s \leq t \\ 0 & otherwise \end{cases} \tag{17}$$

where we now define $p(l|t)$ to ensure normalisation over all $L+1$ states, $l = 0, \ldots, L$.

We also show that it is possible to extend our framework to obtain state-of-the-art results. In our definition of Latte Macchiato, the quantity $p(l|t)$ depends only on the token $x_t$, while $p(s|l,t)$ is based only on $x_s$ from the entire sequence. To encourage the latent states to capture temporal dependencies across multiple sub-words, we can use a 1D convolution of size $K$ to compute these probabilities:

$$y_t = \sum_{i=0}^{K} w_i^c x_{t-i} \tag{18}$$

$$p(l|t) = \frac{e^{y_t^\mathsf{T} w_l^q}}{\sum_{j=1}^{L} e^{y_t^\mathsf{T} w_j^q}} \quad p(s|l,t) = \frac{e^{y_s^\mathsf{T} w_l^k}}{\sum_{s=1}^{t} e^{y_s^\mathsf{T} w_l^k}} \tag{19}$$

We observed that performance improves with larger convolution sizes $K$ prompting us to also extend $y_t$ to depend on all previous tokens using a linear recurrent neural network. For our experiments, we used the recurrent gated linear unit (RGLRU) layer introduced by De et al. (2024), which, compared to a convolution, is also input dependent. Note that both the convolution and recurrent layers break positional invariance, thereby eliminating the need for positional encodings in these extensions.

## 5 Experiments

### 5.1 Synthetic Tasks

Linear models are known for being worse than transformers in retrieval capabilities(Arora et al., 2023). Hence we test the capabilities of our model on the synthetic MQAR data (Arora et al., 2023) and compare it with two other linear models and the standard transformer. Figure 6 shows that Latte performs competitively with the transformer and outperforms the other linear models in our training set. In all the experiments, the window size of attention is 128, being smaller than the context length.

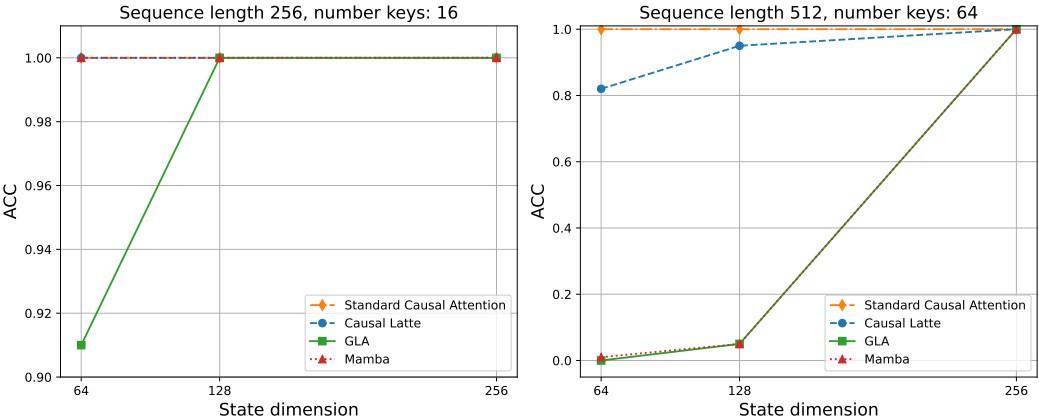

Figure 6: Accuracy (ACC) on MQAR dataset for different sequence lengths and number of key-value pairs. We set the number of test examples to 10000 and train examples to 100000.

### 5.2 Ablations

We train small causal models on OpenWebText (Gokaslan & Cohen, 2019) for next-token prediction using a shared setup across all variants for 8B tokens.

**Base model.** Table 1 reports perplexity across variants. The base model (Latte) uses Latte layers for temporal mixing, with the rest matching a standard Transformer (Vaswani et al., 2017). A "++" variant replaces LayerNorm with RMSNorm (Zhang & Sennrich, 2019) and the feedforward block with a Gated Linear Unit (GLU) (Dauphin et al., 2017), yielding a small performance gain.

**Local attention.** We extend Latte++ with a $K = 3$ convolution (Section 4.1), improving results with minimal parameter overhead. Adding a 128-token sliding window attention (SWA) using ROPE (Su et al., 2024) yields a larger performance boost at the cost of more parameters.

**Recurrent variant.** Replacing the convolution with a Recurrent Gated Linear Unit (RGLRU) further improves performance and slightly reduces parameters. Combining this with SWA produces Latte-R++, our best model.

**Comparison to SOTA.** Table 2 compares Latte to other efficient models. To isolate the effect of Latte, we evaluate R-SWA++, which replaces Latte attention with normalized RGLRU outputs: $Y = \text{RGLRU}(X)$, $Z = \text{RMSNorm}(Y)$, $Q = W_q Z$, $K = ZW_k$, $V = XW_v$, similar to Mega (Ma et al., 2023). Latte consistently achieves the lowest perplexity despite a modest higher parameter count. We keep depth, width, and feed forward dimensions fixed, and match Griffin's SWA window size (128).

Overall, combining global latent attention with local SWA (Latte-Macchiato) achieves performance competitive with state-of-the-art language models. Importantly, Latte can also extend pre-trained models to longer contexts via global attention (Section 5.4).

Table 1: Iterative improvement of Latte language modelling. SWA: Sliding Window Attention.

| Model | Params. | PPL ↓ |
|---|---|---|
| Latte | 111M | 21.88 |
| Latte++ | 140M | 21.56 |
| Latte-Conv++ | 140M | 20.26 |
| Latte-Conv-SWA++ | 153M | 18.52 |
| Latte-R++ | 139M | 19.99 |
| Latte-R-SWA++ | 153M | **17.64** |
| Transformer++ | 151M | **17.19** |

Table 2: Comparison of Latte-Macch with other linear-scaling models on language modelling. SWA: Sliding Window Attention

| Model | Params. | PPL ↓ |
|---|---|---|
| Mega Ma et al. (2023) | 153M | 23.75 |
| Retnet Sun et al. (2023) | 197M | 21.59 |
| H3 Fu et al. (2023) | 125M | 21.0 |
| RWKV Peng et al. (2023a) | 153M | 18.97 |
| Griffin De et al. (2024) | 139M | 18.83 |
| R-SWA++ (our) | 141M | 18.25 |
| Mamba Gu & Dao (2023) | 149M | 17.70 |
| GLA Yang et al. (2024) | 206M | 19.10 |
| Ligth.Att Qin et al. (2024a) | 166M | 23.67 |
| GatedDeltaNet Yang et al. (2025) | 190M | 21.41 |
| Latte-R-SWA++ | 153M | **17.64** |

**Latent Collapse.** A common issue with latent variable models is latent collapse, where only a small subset of latent states is used. As demonstrated in Figure 7, Latte does not exhibit latent collapse, even in the absence of dropout. The figure also highlights that local attention and Latte attention are effectively used, with the probability mass distributed across various latent states. The plots are generated from various heads and layers of the Latte-RGLRU-SWA++ model, as detailed in Table 2. We provide plots for all layers and heads in Figure 7.

### 5.3 Bidirectional Tasks

In our second set of experiments, we evaluate the bidirectional version of Latte on the Long-Range Arena (LRA) benchmark (Tay et al., 2021). We compare it against vanilla linear attention (as introduced by (Katharopoulos et al., 2020)) and other linear attention variants. As shown in Table 3, bidirectional Latte outperforms vanilla linear attention and performs on par with the strongest transformer-based baselines, though it still falls short of state-space models that are time-invariant. When we add a recurrent linear layer, performance improves substantially on the discrete tasks in LRA. However, performance on image-based (continuous) tasks remains weaker. This pattern is consistent with other models like Mamba, which also excel in discrete domains such as language modelling but underperform on continuous data[1]. Finally, performance on discrete tasks improves even further when using a bidirectional sliding window, as in Latte-Macchiato (Latte-R-SWA++).

---

[1]As noted by the authors of Mamba (Gu & Dao, 2023)

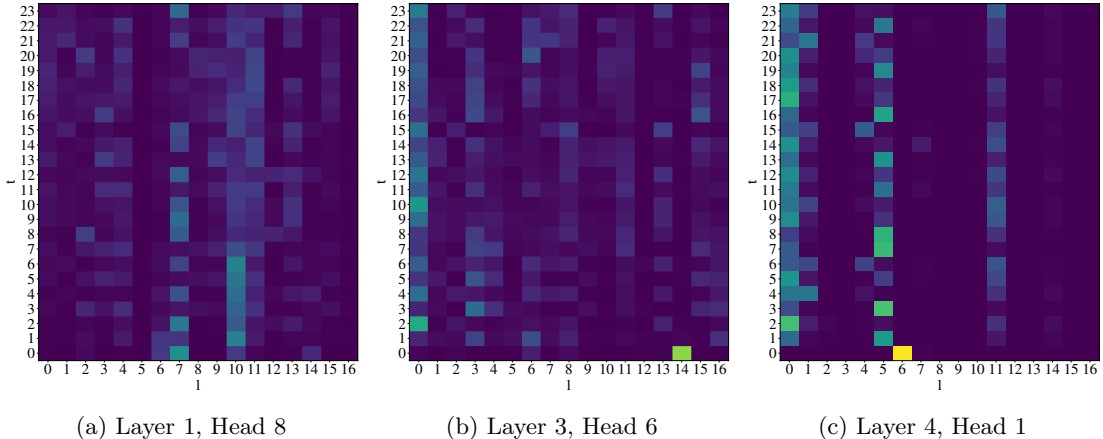

(a) Layer 1, Head 8      (b) Layer 3, Head 6      (c) Layer 4, Head 1

Figure 7: Plots of $p(l|t)$ for different layers and heads across a sequence of 25 tokens and $l = 0, \dots, 16$. State 0 corresponds to using standard causal windowed attention, whereas states higher than zero correspond to global latent tokens. Brighter means higher probability.

Table 3: Classification accuracies for LRA dataset. We report the best test score (higher is better). All Latte versions are bidirectional.

| Model | ListOps | Text | Retrieval | Image | Pathfinder |
|---|---|---|---|---|---|
| Bid. Att. | 36.37 | 64.27 | 57.46 | 42.44 | 71.40 |
| Linformer | 35.70 | 53.94 | 52.27 | 38.56 | 76.34 |
| Longformer | 35.63 | 62.85 | 56.89 | 42.22 | 69.71 |
| Luna Bid. | 38.01 | 65.74 | 79.55 | 47.47 | 78.89 |
| Linear Att. | 16.13 | 65.90 | 53.09 | 42.34 | 75.30 |
| Mega-Chunk | 58.76 | 90.19 | 90.97 | 85.80 | 94.41 |
| S5 | 62.15 | 89.31 | 91.40 | 88.00 | 95.33 |
| Latte | 40.18 | 64.51 | 73.39 | 47.55 | 75.61 |
| Latte-R++ | 56.7 | 83.85 | 81.07 | 57.61 | 72.13 |
| Latte-R-SWA++ | 61.39 | 85.8 | 87.67 | 70.19 | 73.69 |

## 5.4 Extending a Pre-trained Model

Training large models from scratch is computationally expensive, even when the sequence mixing layer (like attention) has linear time and memory complexity. Recent work has demonstrated preliminary success in distilling pre-trained quadratic self-attention layers into sub-quadratic layers, such as Mamba (Bick et al., 2024). However, unlike Latte, these architectures (Gu & Dao, 2023) significantly differ from the attention mechanisms used in standard transformers, making knowledge distillation from pre-trained transformers more complex. Other research has modified relative embeddings in standard attention to enable sequence extrapolation, but the computational cost remains quadratic (Sun et al., 2022). We use Latte-Macchiato with SWA weights taken from a pre-trained large model and show that training only the Latte-specific weights for 1.6B tokens is sufficient. This approach enables us to achieve desirable properties, such as global context and effective sequence length extrapolation, by bootstrapping from a pre-trained open-source large language model. In our experiments, we use a pre-trained 2.6B Gemma model (Gemma-Team, 2024) and replace the standard attention layer with a Latte-Macchiato layer of 128 long sliding window attention. The model is trained on the SlimPajama dataset (Soboleva et al., 2023), for a single day on four 80GB A100 GPUs. In Table 4, we evaluate both the original Gemma model and our modified version, Gemma Macchiato, on the validation set as well as other publicly available corpora[2]. First, on sequences of length 4K, which match the

---

[2]We use the version without copyrighted content

training length, we find that our model's results are comparable to or even exceed those of the original model. When extending the sequence length to 8K and 16K tokens, our model significantly outperforms Gemma, demonstrating that excellent context extrapolation capabilities are acquired with minimal additional training steps.

Table 4: PPL ↓ on the validation set for 4K, 8K and 16K sequences. Gemma-Macchiato is initialised from Gemma and pre-trained on Slim-Pajama (SP) and Tiny-Stories (TS). Unlike Gemma-Macchiato, Gemma fails to generalise to longer sequences.

| | Gemma | | | Gemma-Macchiato | | |
| Data | 4K | 8K | 16K | 4K | 8K | 16K |
|---|---|---|---|---|---|---|
| SP | 10.97 | 36.35 | 294.18 | 10.14 | 9.99 | 10.27 |
| Pile | 7.42 | 19.26 | 243.54 | 7.27 | 6.98 | 7.04 |
| OWT | 10.75 | 38.36 | 252.74 | 10.76 | 10.72 | 10.99 |
| TS | 5.45 | 19.15 | 66.61 | 4.26 | 4.34 | 4.30 |

We also check the abilities of the distilled model on a standard natural language harness of multiple-choice question-answering. Like the general trend of linear models, performance decreases especially on tasks like MMLU (Mercat et al., 2024; Zhang et al., 2024). However, our aim is not to outperform standard attention, but to provide a linear global context extension method which is a better alternative to sequence truncation, often when the quadratic cost of attention becomes a limitation.

Table 5: Common Few Shot learning benchmarks. The score is accuracy or normalized accuracy (↑). We use a sliding window of size 128.

| Model | MMLU | HellaSwag | Lambada | ARC-C | ARC-E | WinoG | Piqa | BoolQA |
|---|---|---|---|---|---|---|---|---|
| Gemma2 2B | 53.0 | 73.03 | 69.8 | 53.4 | 80.2 | 71.4 | 79.1 | 73.61 |
| Gemma-Mach 2B | 46.8 | 73.11 | 68.29 | 52.9 | 76.9 | 70.6 | 78.7 | 71.48 |
| Mamba (3B) | 26.2 | 71.0 | - | 41.7 | 68.2 | 65.9 | 78.1 | 71.0 |
| GLA (1.3B) | - | 49.8 | 46.9 | 26.6 | 57.2 | 53.9 | 71.8 | - |

### 5.5 Empirical Runtime Efficiency

We benchmark the forward-pass runtime of both the convolutional (Latte-Conv-SWA++) and recurrent (Latte-R-SWA++) variants of Latte against standard attention mechanisms across varying sequence lengths and model sizes. All models use identical hyperparameters, with batch size adjusted to keep the total number of tokens constant. As shown in Figure 8b and Figure 8a, both Latte variants outperform standard attention in speed. While we also include Flash Attention as a reference, our JAX-based implementation is not CUDA-optimized, making direct comparison unfair. Nevertheless, at the 2.6B scale, our linear JAX implementation outperforms Flash Attention's CUDA kernel.

Given its stronger performance and modest runtime overhead, the recurrent variant (Latte-R-SWA++) is generally preferable to the convolutional version, likely due to its ability to capture longer-range dependencies. Although our focus has been on reducing asymptotic time and memory complexity, further kernel-level optimizations could improve Latte's practical runtime efficiency.

## Related Work

The literature on efficient attention spans a wide range of techniques, broadly categorized into downsampling (Jaegle et al., 2021), random patterns (Zaheer et al., 2020), learnable patterns (Wang et al., 2022; Kitaev et al., 2019), sparsity (Ainslie et al., 2020; Beltagy et al., 2020), recurrence (Dai et al., 2019), and low-rank approximations (Wang et al., 2020; Katharopoulos et al., 2020). Some recent efforts also exploit hardware parallelism (Qin et al., 2024b; Sun et al., 2024). For a comprehensive survey, see (Lin et al., 2021).

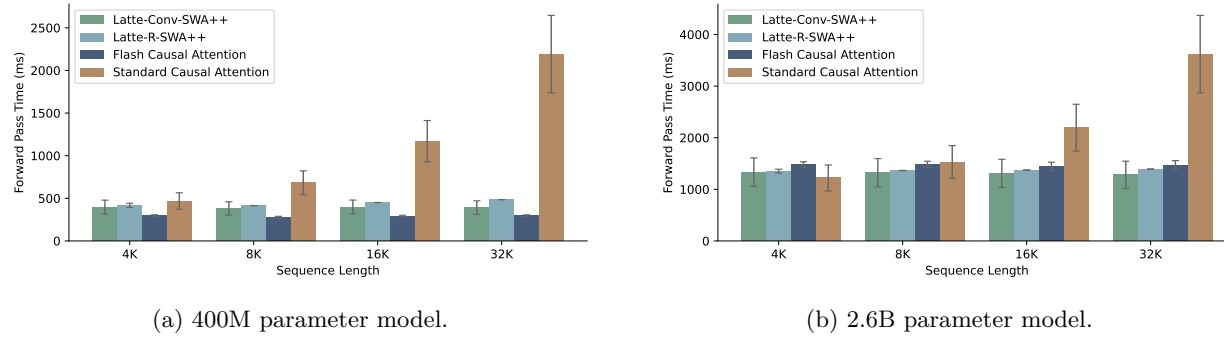

(a) 400M parameter model.  (b) 2.6B parameter model.

Figure 8: Runtime in milliseconds (ms) for forward passes at different sequence lengths for (a) 400M and (b) 2.6B parameter models. Standard deviations for Latte-R-SWA++ are included but too small to be visible.

**Efficient Attention.** Shen et al. (2021) introduced a linear-time bidirectional attention mechanism similar in form to our bidirectional Latte. However, their approach lacks a latent-variable interpretation and does not extend to causal attention. Moreover, their focus is on vision tasks, whereas we target language modelling and explicitly integrate both global and local attention mechanisms.

**Luna.** Luna (Ma et al., 2021) performs attention between input tokens and latent tokens in a bidirectional setting. While Luna shares our use of latent structures, it differs significantly in architecture and parameterisation, especially in the causal setting. Unlike Latte, Luna does not incorporate local attention, making it less effective for capturing both short- and long-range dependencies in language.

**State-space and Hybrid Models.** State-based models (Gu et al., 2021; Smith et al., 2022) replace attention with structured recurrence. Gated variants like Mamba (Gu & Dao, 2023) improve performance on discrete data by introducing input-conditioned dynamics. Hybrid models such as Mega (Ma et al., 2023), Griffin (De et al., 2024), and SSM-Trans (Zuo et al., 2022) combine local attention with recurrent layers. Our method falls into this hybrid category but uniquely combines latent-variable-based global attention with efficient local context through sliding windows. Additionally, Latte-Macchiato enables seamless extension of pre-trained language models to long contexts with linear complexity, a capability not present in most prior work.

## 6  Limitations

Our framework does not cover all linear-time architectures, in particular, sparse-attention methods and non-normalized recurrent sequence mixers (e.g., Mamba) fall outside its probabilistic assumptions. Nevertheless, this perspective enables a new *directed* linear parameterisation that preserves linear time while aligning with causal sequence modelling. Finally, because our method approximates full attention, careful validation and stress testing are required before deployment in high-stakes settings such as healthcare or legal applications.

## 7  Conclusion

We introduced a latent-variable formulation of attention that scales linearly with sequence length and can serve as a drop-in replacement for standard attention. While prior work has explored low-rank approximations, our approach is, to the best of our knowledge, the first to reinterpret linear attention as a latent graphical model. This perspective enables a principled derivation of both bidirectional and causal variants within a unified framework, leading to strong empirical results in language modeling.

Beyond this core formulation, our framework naturally integrates local sliding window attention with global latent attention. This hybrid structure not only improves performance but also offers a practical advantage: it enables the extension of pre-trained language models to much longer contexts with minimal additional training and runtime overhead.

Our experiments focused on language modeling and long-range classification, but the framework is broadly applicable. Future work will explore extensions to tasks such as question answering, sequence-to-sequence generation, and multimodal modeling.

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

## A    Experimental Details

This section describes in detail the datasets and hyperparameters used for our language modelling and classification experiments. For all our experiments we use 4 A100 GPUs.

### A.1    Language Modelling

OpenWebText (Gokaslan & Cohen, 2019) is an open-source version of the corpus used to train GPT (Radford et al., 2019) and consists of 41 GB of data extracted from 8,013,769 documents. We tokenize the corpus using a pre-trained Byte Pair Encoding (BPE) tokenizer with a vocabulary of 50,267 tokens. We also ensure that sequences are consistently of length 1024 by concatenating consecutive tokenized examples until this length is reached. This eliminates the need for padding, ensuring that it is not a confounding factor and results in a more efficient computation.

### A.1.1 Hyperparameters

In this section, we describe the hyper-parameters used in all of the language modelling experiments. Where the hyper-parameter is missing, it means that we vary it in the experiment and its value is clear from the corresponding section in the paper.

Table 6: List of hyperparameters used in the language generation task.

| Hyperparameter | Value |
|---|---|
| $\#Layers$ | 8 |
| $\#Heads$ | 8 |
| Hidden Dim ($D$) | 512 |
| Feed Forward Dim. | 2048 |
| Latent Dim ($L$) | 256 |
| Local Attention Window | 128 |
| Convolution Kernel ($K$) | 3 |
| Dropout | 0.1 |
| $LR$ | $5 \times 10^{-4}$ |
| $LR$-Warmup | 4000 steps |
| $LR$-Decay | Linear |
| $\#Iters.$ | 200K |
| Weight Decay | 0.01 |
| Seq. Len. ($T$) | 512 |
| Batch Size ($B$) | 64 |
| Tokenizer | BPE |
| Embedding Type | Learned |
| Unroll Factor | 32 |

Table 7: Hyperparameters for adopting Gemma to our framework. All the Gemma hyperparameters are kept intact. $LR$ is the learning rate and "$\#$" denotes "the number of".

| HyperParam. | Value |
|---|---|
| Local Attention Window | 128 |
| Latent Dim ($L$) | 128 |
| $LR$ | 0.0006 |
| $LR$-Warmup | 2000 |
| $LR$-Decay | Cosine |
| $\#Iters.$ | 100000 |
| Seq. Len. ($T$) | 4096 |
| Batch Size ($B$) | 4 |
| Tokenizer | Gemma2 |

### A.2 LRA Dataset

This section displays the hyperparameters employed in the bidirectional experiments and provides a brief description of the synthetic datasets utilized in the LRA corpus. A more comprehensive account can be found in the original paper (Tay et al., 2021).

In the experiments, one layer consists of a standard transformer block where the transformation operation that gives $\tilde{x}_t$ is Latte or Standard Attention. For positional encoding, we use the classic transformer sinusoidal embeddings. This convention holds for both bidirectional and unidirectional problems. A complete implementation can be found in our code repository: `https://github.com/raresdolga/latte_transformer`.

Table 8: Hyperprameters used for training on LRA. Number of latent states $L$ specified in the result table. $H$=number heads, $D$=hidden dimension, $LR$=learning rate, $B$=batch size, $WD$=weight decay. $\#Layers$ denotes the number of layers which include attention/approximation of attention and non-linear projections. "Embed." is the type of embedding used by the SWA.

| Dataset | $\#Layers$ | $H$ | L | $D$ | $LR$ | $B$ | $WD$ | Dropout | Epochs | Embed. |
|---------|-----------|-----|-----|-----|------|-----|------|---------|--------|--------|
| ListOps | 6 | 4 | 40 | 128 | 1e-3 | 64 | 0.01 | 0.1 | 50 | Rope |
| Text | 6 | 4 | 256 | 256 | 1e-3 | 32 | 0.05 | 0.1 | 32 | Rope |
| Retrieval | 6 | 4 | 40 | 128 | 1e-4 | 32 | 0.01 | 0.1 | 20 | Rope |
| Image | 6 | 4 | 40 | 512 | 1e-3 | 32 | 0.05 | 0.1 | 200 | Absolute |
| Pathfinder | 6 | 4 | 256 | 256 | 1e-3 | 64 | 0.03 | 0.2 | 200 | Absolute |

### A.2.1 ListOps

The dataset contains sequences up to length 2048 of numbers from 0 to 9 and four operators: *MAX, MEAN, MEDIAN* and *SUM_MOD*. Parentheses are used to delimit the reach of each operator. The answer is also a digit from 0 to 9, which allows us to easily transform the problem into a ten-way classification task.

### A.2.2 Text

The Text corpus is represented by a binary classification task with long text sequences. One can easily obtain large contexts from existent datasets by tokenizing at the character level. This part of the benchmark is derived from the IMDb (Maas et al., 2011) movie review corpus, resulting in 4K character sequences.

### A.2.3 Retrieval

This dataset tests the ability of a model to predict the similarity between two long documents. Similarly to the previous corpus, it ensures long contexts through character-level tokenization, resulting in 4K tokens per document. Using a "two-tower model setup" (Tay et al., 2021) the total sequence length becomes 8K. This is a binary classification problem, which uses accuracy as a metric.

### A.2.4 Image

Alongside text, images can also exhibit long-range dependencies by flattening the original image into a sequence. The Image dataset is the sequential version of Cifar10 (Krizhevsky et al., 2009), which contains images of 10 different entities: "airplane, automobile, bird, cat, deer, dog, frog, horse, ship, truck". To obtain a sequence with one input channel we apply a grayscale transformation. The model needs to predict the correct entity class, given the flattened image represented as a sequence of tokens.

### A.2.5 PathFinder

This part of the benchmark is also represented by images where the task is to predict whether there is a path between two points in a black-and-white image. This dataset consists of $32 \times 32$ images which after flattening result in sequences of length 1024. In general larger sequences can be created by increasing the resolution. Data is tokenized similarly to the image dataset in Section A.2.4.

### A.2.6 PathX

This dataset is a version of PathFinder where the image size is increased to $128 \times 128$, resulting in flattened sequences of 16384 tokens. Since all the transformer architectures fail on this dataset, we do not add it to the benchmark.

# B   Additional Experiments

## B.1   Length Extrapolation

**Sequence Extrapolation.**   Considering that long sequences improve perplexity, we also examine the model's ability to extrapolate to sequences longer than those seen during training. Specifically, we train on 5K-token sequences from the BookCorpus dataset and evaluate on sequences up to 16K tokens. As shown in Figure 9, our model successfully extrapolates to longer sequences, whereas the performance of standard causal attention degrades as the sequence length increases. Notably, Latte-RGLRU-SWA++ also achieves performance comparable to standard attention on sequences seen during training. We use YARN (Peng et al., 2023b) relative positional encoding for standard attention and ROPE for sliding window attention in Latte. YARN is used because it helps transformers extrapolate.

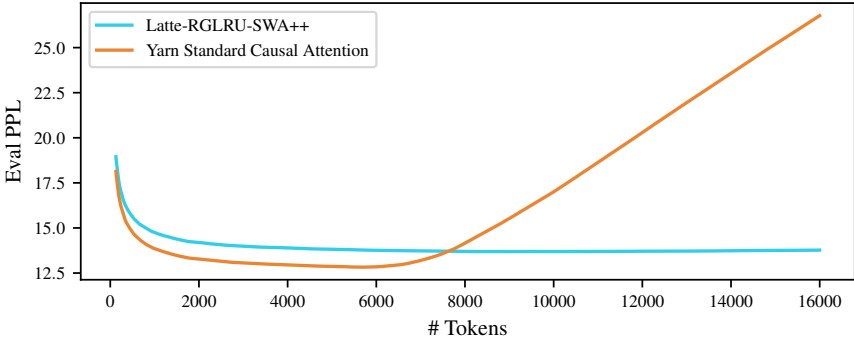

Figure 9: Sequence Length Extrapolation for Attention and Latte-Macch. Both models were trained on sequences of length 5K which are extrapolated to 16K during testing.

## C  Causal Latte Implementation

Listing 1: Scan version of Latte.

```
@partial(jax.jit, static_argnums=(3, 5))
def causal_latte(Wq, Wk, Wv, H, X, unroll=100):
    """
    Scan implementation of latte.
    B: batch size H: nr heads, T: seq_len, D: hidden_dim. L: number latent states
    Args:
        Wq: jnp.array(DL), Wk:jnp.array(DL), Wv:jnp.array(DM) - parameter matrices
        H: int - nr heads
        X: jnp.array(BTD) - input
        unroll: int - unroll of the loop
    Returns:
        y: jnp.array(BTD) - transformed output sequence
    """
    def accumulate(carry, args):
        csum, norm_cumsum, prev_mx = carry
        Qs_t, curr_alph, V_t, c_mx = args
        revert_maxi = jnp.exp(-c_mx + prev_mx)
        add_maxi = jnp.exp(curr_alph - c_mx)
        norm_cumsum = jnp.einsum("BHL,BHL->BHL", norm_cumsum, revert_maxi)
        norm_cumsum += add_maxi
        carry = jnp.einsum("BHLD,BHL->BHLD", csum, revert_maxi)
        carry += jnp.einsum("BHL,BHD->BHLD", add_maxi, V_t)
        y = jnp.einsum("BHL,BHLD->BHD", Qs_t / norm_cumsum, carry)
        return ((carry, norm_cumsum, c_mx), y)

    B, T, D = X.shape
    L = Wk.shape[-1]
    V = jnp.einsum("DM,BTD->TBM", Wv, X).reshape(T, B, H, -1)
    Q = jnp.einsum("DL,BTD->TBL", Wq, X).reshape(T, B, H, -1)
    K = jnp.einsum("DL,BTD->TBL", Wk, X).reshape(T, B, H, -1)
    maxi = jax.lax.cummax(K, axis=0)
    init_alpha = jnp.zeros(shape=(B, H, L // H))
    init_carry = jnp.zeros((B, H, L // H, D // H))
    Qs = jax.nn.softmax(Q, axis=-1)
    _, y = jax.lax.scan(
        accumulate,
        unroll=unroll,
        init=(init_carry, init_alpha, K[0]),
        xs=[Qs, K, V, maxi],
    )
    y = y.transpose(1, 0, 2, 3)
    return y.reshape(B, T, D)
```

