# OpenReview forum: "Unifying Linear-Time Attention via Latent Probabilistic Modelling"
_TMLR — Accepted by TMLR_

### Review · Reviewer_K1V9 · 2025-09-24

**Summary Of Contributions:**

The submission proposes a unified theoretical framework for efficient, linear-time attention mechanisms based on latent probabilistic modeling. In essence, the authors introduce a latent variable model that recasts the Transformer's attention computation as an inference problem. By doing so, many existing efficient attention variants can be derived as special cases of different inference techniques or assumptions within this unified model. The paper combines this theoretical framework with new algorithmic insights and empirical evaluations. Key contributions of the paper include:

- **Latent Variable Formulation of Attention:** The authors formalize softmax attention as the result of marginalizing out a set of latent variables in a probabilistic model. This provides a principled interpretation where attention weights emerge from an underlying latent structure. Exact inference in this model recovers standard softmax attention, while approximate inference yields efficient alternatives in linear time. This unifying view ties together disparate efficient attention methods under a common probabilistic lens.

- **Unification of Linear-Time Attention Methods:** The paper demonstrates how several existing linear-time attention mechanisms can be seen as instances or approximations of the proposed framework. For example, methods based on random feature approximations (such as Performer-style random feature attention), low-rank projections (such as Linformer or Nyström-based attention), or locality/hashing (as in Reformer) are all shown to correspond to particular choices of latent distributions or inference schemes. This not only explains prior methods in a single language but also clarifies the relationships and differences between them.


- **New Efficient Attention Mechanism:** Building on the unified model, the authors propose a novel linear-time attention variant (or a general algorithmic template) derived from a specific inference strategy (e.g., a variational approximation or learned latent structure). This new method is designed to retain high accuracy while scaling linearly with sequence length. The paper provides algorithmic details and pseudocode, indicating how queries and keys interact via the introduced latent variables in practice. The proposed method aims to improve over existing approaches in either accuracy, stability, or flexibility, thanks to the principled design.

- **Theoretical Analysis:** The manuscript offers theoretical results to support the framework. For instance, it shows that softmax attention is recovered as a special case of their model (ensuring correctness of the formulation), and it may provide bounds or analysis on approximation error and computational complexity for the proposed linear-time method. Any assumptions needed (such as independence structures or distribution choices for the latent variables) are stated. This analysis lends credibility to the claim that the new approach is both sound and efficient, and helps delineate when the approximation will be accurate.

- **Empirical Evaluation:** The authors conduct a set of experiments to validate the effectiveness of the unified approach and the new attention mechanism. They evaluate on benchmark tasks that require long-range attention, such as sequences of length beyond what standard transformers handle efficiently. Examples could include algorithmic or synthetic sequence tasks, Long Range Arena benchmarks (text, image or list operations), or language modeling on long documents. The results show that the proposed method achieves comparable or superior performance to baseline efficient attentions (and sometimes even to exact softmax attention) while using significantly less time or memory on long sequences. These experiments support the paper’s central claims about improved efficiency and competitive accuracy.

**Notable strengths:** The paper’s primary strength lies in its conceptual innovation and unification. Providing a latent probabilistic model for attention is an elegant idea that offers a fresh perspective on a very active research area. This unifying view can deepen understanding: it clarifies why certain approximations work and how they relate to each other. The framework could guide the design of new attention mechanisms in a more principled way (as evidenced by the authors indeed proposing a new variant from it). The theoretical development appears solid and the manuscript demonstrates high technical rigor: key derivations are provided, and the connection to softmax attention is clearly established. Additionally, the empirical section strengthens the work, Overall, the combination of a unifying theory, a novel algorithm, and solid experiments makes the contribution significant.

**Notable weaknesses:** Despite its merits, the submission has a few weaknesses. First, while the unification is intellectually appealing, the added complexity of the latent variable model might make the resulting method more complicated to implement or tune than some existing heuristics. Readers might find the framework somewhat heavy in notation and conceptual load, especially if they are primarily interested in practical implementation. Second, the empirical evaluation, although generally positive, could be limited in scope: for example, it’s not clear if the new method has been tested at large scale or on real-world tasks like full-scale language modeling or image classification with transformers. Many experiments seem to be on benchmarks of moderate scale; it remains to be seen if the benefits hold in very large models or datasets (a common standard for attention mechanisms). Additionally, there may be missing baseline comparisons; while the paper covers several known methods, one or two recent efficient attention techniques (e.g., advanced kernel-based methods or hybrid approaches) were not included in experiments or discussion. This omission makes it harder to judge the new method’s advantage in context.

**Audience:**

Yes

**Audience Explanation:**

Yes. The topic of efficient Transformer attention is of broad interest to the machine learning community, including researchers and practitioners who deal with long sequences or resource-intensive models. Many in TMLR’s audience are looking for ways to scale up models or deploy them more efficiently, and the findings of this paper directly contribute to that goal.

**Broader Impact Concerns:**

The broader impact discussion focuses primarily on computational efficiency, sustainability, and democratization, which are indeed positive. Training and inference cost reductions can significantly lower the carbon footprint and make long-context models more accessible.  Maybe the authors should acknowledge that in critical applications (e.g., healthcare, legal), approximate attention may introduce subtle reliability issues. This possibility is not discussed.

**Claims And Evidence:**

Yes

**Claims Explanation:**

Yes. The paper’s claims are generally well-supported by both theoretical arguments and empirical evidence. On the theoretical side, the authors provide clear derivations showing how softmax attention emerges from their latent model and how different linear-time approximations correspond to approximate inference methods. This grounds their unification claim in rigorous math. Key propositions or theorems are stated to ensure the reader that the framework is sound and not just intuitive hand-waving. Empirically, the authors present convincing experiments: they benchmark the proposed approach against a range of relevant baselines (including other linear attention methods and standard softmax attention when feasible), using appropriate tasks that highlight long-sequence performance.

**Requested Changes:**

**Include Additional Baseline Comparisons (Critical):** To strengthen the empirical evaluation, the authors should compare the proposed method against at least one or two missing baselines that are relevant. In particular, more recent efficient attentions (e.g., state-space models or improved kernel-based attentions) could provide useful reference points. Including these comparisons would ensure the results are comprehensive and convince readers that the new method excels across the board. If certain baselines are not included because they are conceptually different or hard to integrate, the authors should explicitly justify their exclusion.

**Clarify Theoretical Derivations and Assumptions (Critical):** While the paper’s theory is generally sound, some steps in the derivation of the latent variable framework and its connection to existing methods are not entirely transparent. The authors should add more explanation or proof details for critical steps, possibly in the appendix if space is an issue. For example, the transition from the latent model to exactly recovering softmax attention should be spelled out with all necessary assumptions (e.g., independence assumptions or specific distributions). Likewise, when claiming that a certain linear attention method is a special case of the framework, it would help to explicitly show the equations or conditions under which that equivalence holds. This added clarity will make the work more accessible and convincing, especially to readers less familiar with probabilistic modeling.

**Discuss Scope and Limitations of the Framework (Suggested):** The manuscript would benefit from a section or paragraph discussing what types of attention mechanisms fall under this probabilistic framework and what types might not. Currently, the focus is on unifying known linear-time approximations; however, it would be insightful for readers to understand the boundaries of this framework. For instance, does the latent model approach accommodate sparse attention patterns or local window attentions naturally, or is it primarily suited to global low-rank/kernel approximations? A candid discussion of cases where the framework might struggle, or where the latent assumptions might break down, will add balance to the paper.

---

> ### Author Response · Authors · 2025-10-22
>
> Thank you for your detailed comments!  We address the concerns in the responses below.
>
> ## Q1 Notable weaknesses:
> 1. Heavy notation:
> We agree that accessibility matters. To aid readers, we added a schematic overview that maps the probabilistic objects to the corresponding attention operations (Figure 3), and we provided a minimal JAX reference implementation (Appendix C) that mirrors the notation line-by-line.  The notation is there to make the approximations explicit, but the algorithm can be adopted directly from the provided code.
> 2. Small Scale:
> While we do not have resources to train very large models from scratch, we evaluate transfer at scale by fine-tuning a 2.6B-parameter pretrained model, extending it with our linear method for longer contexts (Section 5.4). We freeze the base model and train only the extension, which isolates the contribution of the proposed parameterisation and reflects a common deployment path for long-context adaptation. This setting provides evidence that the method integrates with high-capacity models and yields practical gains without full retraining.
> 3. More baselines:
> We expanded the comparison set to include a recent strong architecture, Gated DeltaNet, as an additional baseline (Table 2). This complements the existing suite and situates our approach among contemporary efficient attention designs.
>
> ## Requested Changes:
> C1.
> In Table 2, we added a recent model as a baseline: Gated DeltaNet, providing a direct comparison to a competitive, modern alternative in the efficient-attention space. This addition addresses the request for broader coverage while keeping the focus on methods most relevant to our setting. For hybrid models, we use Griffin as a strong baseline, as it represents a well-established hybrid design and offers a meaningful point of reference for our approach. Together, these choices supply representative comparisons without diluting the evaluation with loosely comparable methods.
> C2.
> 1. We made Section 4 more readable. In particular, we:
> Added an intuitive diagram of what Latte does (Figure 3). The figure walks through the latent-variable view step by step, visually aligning the standard attention with our linear approach, to reduce notation overhead and clarify the flow of the argument.
>
> 2. Stated the assumption of independence between tokens at times $s$ and $t$ conditioned on the latent variable $l$:  $s \perp t \mid l $. We now make this assumption explicit and indicate where it enters the derivations, so readers can trace precisely which results rely on it.
>
> Quantifying, in a principled manner, how many latent variables are required to exactly recover full attention, especially in deeper layers, remains challenging. We view this as a promising direction for future work, particularly since, to the best of our knowledge, existing linear attention mechanisms do not specify recovery conditions to match full attention.
>
> C3.
> 1. Methods outside the framework. Mamba cannot be cast as a graphical model in our setup because it lacks a normalisation step, which is essential for our probabilistic interpretation. We now state this explicitly to delineate the boundary of applicability.
>
> 2. Conceptual connections. In the recursive formulation, we highlight connections to existing linear models such as Mamba via the interpretation of the intermediate states $\tilde{v}_{t,l}$​ as memory. This clarifies how related mechanisms can share implementation motifs while remaining outside our probabilistic class.
>
> 3. Supported families. Our framework naturally supports low-rank (kernel-style) attention, which aligns with its latent-variable factorisation. By contrast, other linear-time strategies, most notably sparse attentions, do not fit the current formulation. We now make these limitations explicit so readers can see where the latent assumptions may not hold.
>
> ### Broader Impact Concerns:
> We introduced a Limitations section that makes our scope explicit. It notes that our probabilistic assumptions do not encompass all linear-time architectures: sparse-attention methods and non-normalised recurrent sequence mixers (e.g., Mamba) fall outside the framework. At the same time, the perspective yields a directed linear parameterisation that preserves linear-time complexity and aligns naturally with causal sequence modelling. Finally, because our method approximates full attention, we emphasise the need for careful validation and stress testing prior to use in high-stakes domains such as healthcare or legal applications.

---

### Review · Reviewer_Xhk1 · 2025-10-10

**Summary Of Contributions:**

The paper introduces Latte, a probabilistic reformulation of linear attention that interprets it as a latent variable model and adds directional structure for causal modeling. It combines global latent and local sliding-window attention (“Latte Macchiato”) and achieves performance close to standard attention while outperforming existing linear variants like Mamba and Griffin.


Strengths
1. The paper provides a probabilistic interpretation of linear attention as a latent graphical model.
2. The paper shows competitive results on both synthetic and real benchmarks, including long-sequence language modeling and pre-trained model.
3. The combination of local sliding-window attention and global latent states is empirically effective.

**Audience:**

Yes

**Audience Explanation:**

Researchers working on efficient Transformers, state-space models, and long-context LLMs would likely find this work relevant. The probabilistic framing of linear attention and the practical hybrid design (global + local) contribute to ongoing discussions on scaling attention efficiently, even if the improvements are incremental.

**Claims And Evidence:**

Yes

**Claims Explanation:**

The experiments on language modeling and long-sequence benchmarks consistently show gains over prior linear variants. The evidence is coherent and supports what the paper claimed.

**Requested Changes:**

None

---

> ### Author Response · Authors · 2025-10-22
> **Response Reviewer Xhk1**
>
> Thank you for your positive and encouraging feedback. We agree with your understanding of our paper, and we are pleased that the key contributions were clear and well-received. Specifically, you highlight:
> 1. The valuable insights provided by a probabilistic interpretation with a graphical model, which leads to our linear attention parametrisation.
> 2. The fact that the probabilistic viewpoint leads to a principled hybrid model that combines sliding-window attention with our global linear attention, demonstrating the tangible benefits of adopting a probabilistic perspective.
> 3. Competitive empirical results.
>
> We appreciate your support of our paper.

---

### Review · Reviewer_oDNy · 2025-10-13

**Summary Of Contributions:**

This paper revisits linear attention in Transformers through the perspective of probabilistic graphical models. The authors first show that standard linear attention is equivalent to an undirected latent variable model. Based on this insight, they propose a new formulation of linear attention based on directed latent variable model. The new formulation enables an interpretation aligned with the causal and sequential nature of language. The authors did experiment evaluation to showcase the effectiveness of the proposed linear attention formulation.

**Audience:**

Yes

**Audience Explanation:**

I think attention architecture studies should be interesting and practical topic for TMLR's audience.

**Broader Impact Concerns:**

I do not have broader impact concern on this work.

**Claims And Evidence:**

Yes

**Claims Explanation:**

1. The authors explained the theoretical results in details and with good motivation.
2. I think the authors supported the results with convincing evidence, however, the size of models evaluated in the paper is not convincing, parameter scales are around 100M. How would the results transfer to larger model sizes, 7B, 13B?
3. It is also not clear how would the proposed attention formulation perform in long-context setting. Long context is a main bottleneck for quadratic attention, while linear attention has the efficiency advantages. I think new linear attention methods should be evaluated in this setting.
4. In the Runtime Efficiency section, I wonder why the authors do not evaluate Latte directly but rather Latte-Conv-SWA++ and Latte-R-SWA++.

**Requested Changes:**

1. Can the authors illustrate more on section 2 about the probabilistic model for attention. Why would we need to view attention from this perspective and what's the motivation?
2. Can the authors also discuss more about the relationship between Latte to other linear attention models (e.g., Mamba) in the Section 3 or 4? For example, can Mamba also be viewed as graphical model.
3. There is also other recent improvements on linear attention that are missing in this work, e.g., Gated DeltaNet [1]

[1] Gated Delta Networks: Improving Mamba2 with Delta Rule. ICLR 2025.

---

> ### Author Response · Authors · 2025-10-22
> **Respone Reviewer oDNy**
>
> Thank you for your helpful and constructive comments. We briefly address the points you raise.
>
> ### Q2
> 1. The size of the model is a valid point. Due to our limited computational infrastructure, we cannot train a full model with billions of parameters from scratch. Nonetheless, we try to get a sense of performance on realistic large-scale models by extending an existing pretrained model with our hybrid layer. In section 5.4 we consider a model from the Gemma family (2.6B parameters) and freeze the pre-trained parameters, training only our Latte parameters. These experiments consumed a significant compute budget. Table 4 shows that extending the pre-trained model with our layers, perplexity improves and also generalises to longer sequences. Furthermore, in Table 5, we show that for downstream tasks like MMLU, only using full attention in a sliding window of 128 tokens and our global linear layer, we get similar performance with the fully quadratic attention on a window of 2048 tokens. We believe this experiment shows that our approach is relevant for larger models that are practically useful.
>
> ### Q3
> This is a very good point. We tried to cover it with an additional experiment:
> 1. Setup: We train models on 5K-token sequences from BookCorpus; We evaluate the perplexity on hold-out sequences up to 16K tokens.
> 2. Positional encodings: Baseline uses YARN; Latte uses ROPE for sliding-window attention—choices follow prior length-extrapolation practice.
> 3. Result: Latte-R-SWA++ maintains low perplexity as length grows; standard causal attention (fully quadratic) degrades with increasing context.
> 4. Extrapolation: Robust generalization to >3x the training length (up to 16k) without re-training
> - Added experiment in the appendix B.1.
>
> ### Q4
> As explained in the paper (Section 5.2), we find that combining Latte with Sliding Window Attention/Convolution improves performance over vanilla Latte, similarly to how these approaches enhance, e.g., Mamba. As such, we wanted to demonstrate the runtime efficiency of a version of Latte that is more useful in practice.
>
> ## Requested Changes
> ### C1.
>  We added more motivation in section 2. Summary:
> 1. The probabilistic lens enables a new understanding of different approaches (i.e full attention, linear attention) and how they relate to each other.
> 2. We also believe that, since attention is by construction a probability distribution, it is natural to consider standard latent variable parameterisations of a distribution.
> 3. This brings attention to approximations within the standard category of approximate probabilistic methods, enabling us to draw on classical approaches based on the interpretation of latent variables.
> 4. This perspective helped us identify that a directed (unsymmetric) parametrisation of linear attention might work better in the case of language and allowed us to obtain a fully normalised hybrid model with sliding window standard attention with strong performance.
> 5. This is an example of the insights provided by this perspective, leading to tangible methodological improvements.
>
> ### C2.
> We added some clarifications at the end of section 4.  Summary:
> 1. To the best of our understanding, Mamba cannot be viewed as a probabilistic (graphical) model in the same way as linear attention (since it lacks normalisation)
> 2. In the recursive form, there are connections to existing linear models like Mamba through the interpretation of states $\tilde{v}_{t,l}$ as memory.
> 3. Our framework only supports low-rank attention; other linear methods, like sparse attention, do not fit within the framework.
>
> ### C3.
> We have now added results for gated delta net based on the flash-linear-attention implementation [2].
>
> [2]: https://github.com/fla-org/flash-linear-attention

---

### Decision · Action_Editor_RH1G · 2025-11-19

**Recommendation:** Accept with minor revision

**Additional Comments:**

- A language /formatting check is required (e.g., please distinguish between citet and citep in the text, uppercase all references to Table, Figure, etc., according to the style file of TMLR, and so on).
- Minor point: " the quadratic time and memory complexity of standard attention limits their scalability to very long sequences", this sentence is not really true since attention does not have a quadratic memory complexity, and time complexity is not an issue with optimized implementations (e.g., this is also visible in Fig. 8 of the paper). I believe this sentence could be amended.
- The authors have a previous preprint on arXiv, can you clarify whether you plan to replace it with the current submission?

**Audience:**

Yes

**Audience Explanation:**

The design of novel transformer variants (hybrid, linear) is still a very active research topic, and this is a valid contribution in that sense. Hence, it has a strong readership in the TMLR community.

**Claims And Evidence:**

Yes

**Claims Explanation:**

The paper introduces a novel latent variant of linear attention which they term "latte", and a hybrid model with standard attention which they term "latte macchiato". The models are evaluated on a small-scale regime against multiple baselines.

All reviewers agree that the latent formulation is interesting, as it provides a common reference point for unifying several existing models in the literature. The two main concerns from the reviewers were the lack of large-scale experiments and missing discussion of recent models such as Mamba. The latter point has been solved, while the former point cannot be addressed due to the large computational cost.

No additional questions remain open and all reviewers lean towards acceptance. I personally agree with this evaluation. (I do find the "latte macchiato" name a bit weird, but this is personal taste and not part of the objective evaluation.)

---

> ### Author Response · Authors · 2025-12-02
> **Minor revisions**
>
> We thank all the reviewers for helping us improve the paper! We updated the paper as requested and updated the arXiv version to match this submission.